# Predictors of Ascending Aorta Enlargement and Valvular Dysfunction Progression in Patients with Bicuspid Aortic Valve

**DOI:** 10.3390/jcm10225264

**Published:** 2021-11-12

**Authors:** Angela Lopez, Ilaria Dentamaro, Laura Galian, Francisco Calvo, Josep M. Alegret, Violeta Sanchez, Rodolfo Citro, Antonella Moreo, Fabio Chirillo, Paolo Colonna, María Celeste Carrero, Eduardo Bossone, Sergio Moral, Augusto Sao-Aviles, Laura Gutiérrez, Gisela Teixido-Tura, Jose Rodríguez-Palomares, Arturo Evangelista

**Affiliations:** 1Department of Cardiology, University Hospital Vall d’Hebron, CIBERCV, 08035 Barcelona, Spain; kelals@hotmail.com (A.L.); ilaria.dentamaro@gmail.com (I.D.); lauragaliangay@gmail.com (L.G.); saoavilesaugusto@gmail.com (A.S.-A.); lauraguga@gmail.com (L.G.); gisela.tt@gmail.com (G.T.-T.); jfrodriguezpalomares@gmail.com (J.R.-P.); 2Cardiology Department, Hospital Alvaro Cunqueiro, 36213 Vigo, Spain; fcalvoi@me.com; 3Cardiology Department, Hospital Universitari Sant Joan de Reus, IISPV, Universitat Rovira i Virgili, 43204 Reus, Spain; txalegret@hotmail.com; 4Cardiology Department, University Hospital 12 de Octubre, 28041 Madrid, Spain; violetasan@gmail.com; 5Cardiology Department, University Hospital “San Giovanni di Dio e Ruggi d’Aragona”, 84125 Salerno, Italy; rodolfocitro@gmail.com; 6Cardiology Department, Niguarda Ca’ Granda Hospital, 20162 Milan, Italy; antonella.moreo@ospedaleniguarda.it; 7Cardiology Department, Bassano del Grappa General Hospital, 36061 Bassano Del Grappa, Italy; fabio.chirillo@aulss7.veneto.it; 8Cardiology Department, Polyclinic Hospital of Bari, 70124 Bari, Italy; colonna@tiscali.it; 9Instituto Cardiovascular San Isidro, Sanatorio Las Lomas, Buenos Aires 3031, Argentina; dra.celestecarrero@gmail.com; 10Cardiology Department, Azienda Ospedaliera di Rilievo Nazionale Antonio Cardarelli, 80131 Napoli, Italy; eduardo.bossone@aocardarelli.it; 11Servei de Cardiologia, Hospital Josep Trueta, 17007 Girona, Spain; moral.sergio@yahoo.es; 12Heart Institute, Teknon Medical Center-Quirón Salud, 08022 Barcelona, Spain

**Keywords:** bicuspid aortic valve, aneurysm, aortic stenosis, aortic regurgitation

## Abstract

Bicuspid aortic valve (BAV) patients are at high risk of developing progressive aortic valve dysfunction and ascending aorta dilation. However, the progression of the disease is not well defined. We aimed to assess mid-long-term aorta dilation and valve dysfunction progression and their predictors. Patients were referred from cardiac outpatient clinics to the echocardiographic laboratories of 10 tertiary hospitals and followed clinically and by echocardiography for >5 years. Seven hundred and eighteen patients with BAV (median age 47.8 years [IQR 33–62], 69.2% male) were recruited. BAV without raphe was observed in 11.3%. After a median follow-up of 7.2 years [IQR5–8], mean aortic root growth rate was 0.23 ± 0.15 mm/year. On multivariate analysis, rapid aortic root dilation (>0.35 mm/year) was associated with male sex, hypertension, presence of raphe and aortic regurgitation. Annual ascending aorta growth rate was 0.43 ± 0.32 mm/year. Rapid ascending aorta dilation was related only to hypertension. Variables associated with aortic stenosis and regurgitation progression, adjusted by follow-up time, were presence of raphe, hypertension and dyslipidemia and basal valvular dysfunction, respectively. Intrinsic BAV characteristics and cardiovascular risk factors were associated with aorta dilation and valvular dysfunction progression, taking into account the inherent limitations of our study-design. Strict and early control of cardiovascular risk factors is mandatory in BAV patients.

## 1. Introduction

Bicuspid aortic valve (BAV) is the most common congenital heart abnormality with a prevalence of around 1.5% [1]. This condition is associated with a high prevalence of valvular dysfunction and progressive proximal aorta dilation, which may eventually lead to aortic valve surgery and ascending aorta replacement [2,3,4]. Community studies showed that, 20 years after diagnosis, aortic valve surgery or some type of cardiovascular surgery was required in approximately 25% of patients with BAV [5]. Patients with BAV are at high risk of developing aortic valve dysfunction, either stenosis or regurgitation, or both. The distribution of aortic valve dysfunction changed as age increased [6]. In older individuals, the most frequent indication for surgical intervention is aortic stenosis (AS); however, this has been reported to occur around 10 years earlier than in patients with tricuspid aortic valves (TAV) [7]. Several associations among valve morphotypes, cardiovascular risk factors, hemodynamic conditions and the risk of valvular dysfunction and aorta dilation have been addressed in several cross-sectional studies, yielding contradictory data in the different publications [8,9,10,11]. Awareness of these associations would be essential for implementing personalized follow-up, treatment and lifestyle recommendations. 

The present study aimed to assess the mid-long-term progression of aortic dilation and valvular dysfunction in patients with BAV and define the predictors of disease progression.

## 2. Methods

### 2.1. Study Population

This was a retrospective observational study of 718 consecutive patients, over 18 years of age, diagnosed of BAV identified from the echocardiographic database between 2005 and 2015 at 10 tertiary hospitals. Patients were followed for more than 5 years at the cardiac outpatient clinics of those hospitals and demographic information and clinical data were extracted from hospital records.

Patients with aortic coarctation or other congenital disorders, genetic syndromes, previous aortic valvuloplasty, corrective aorta surgery, aortic valve endocarditis, left ventricular dysfunction (EF < 55%), severe valvular dysfunction and ascending aorta dilation >50 mm in the baseline study were excluded. Subjects were censored if they underwent aortic valve or proximal aorta replacement. This retrospective study was approved by the institutional review board of each hospital. 

### 2.2. Echocardiography

Echocardiographic examinations were performed with the use of standard techniques and commercially-available equipment. Echocardiographic parameters were extracted from digital TTE reports under the supervision of an expert at each center. All BAV cases with or without raphe were included in the study. BAV morphotype was categorized as right and left (RL) coronary cusp fusion (anteroposterior BAV), right coronary and non-coronary (RN) cusp fusion (right–left BAV) and left coronary and non-coronary (LN) cusp fusion. Anatomic measurements and valvular dysfunction quantification adhered to the American Society of Echocardiography guidelines and EACVI recommendations [12,13]. Patients with mixed valvular dysfunction were classified according to the predominant functional valve lesion. Significant valvular dysfunction was considered when the degree was more than mild. The degree of valvular calcification was established using the following grading: grade 0 = no evidence of calcification, grade I = localized calcification < 3 mm; grade II = multiple focal calcifications >3 mm; and grade III = extensive valvular calcifications. Calcified aortic valve was considered when grades II and III were visualized.

The ascending aorta was measured by two-dimensional echocardiography using the parasternal long-axis view. Aortic diameter was measured at the aortic root (maximum dilation of Valsalva sinuses) and tubular ascending aorta at the level of the maximum ascending aorta diameter; measurements were taken using the leading edge-to-leading edge convention in end-diastole. Normal aorta size was defined by the reference values reported for the aortic root and ascending aorta based on established guidelines, considering age, body size and sex [14]. Z-score (values adjusted for age, sex and body size) was calculated for both the aortic root and ascending aorta [15]. The aorta phenotype classification included in this study was assigned according to the segment of the vessel with the largest diameters: ‘ascending aorta’ type, if the diameter of the tubular segment exceeded that at the root, and ‘root’ type if the maximum diameter observed was at the level of the sinuses.

Aorta enlargement progression at the aorta root and ascending aorta was analyzed by the annual growth rate defined as the difference between the diameter at the last control and the diameter at the first study divided by the follow-up time interval in years. Predictors associated with rapid aorta diameter progression were analyzed considering the upper quintile of the growth rate variable. Progression of valvular dysfunction severity was assessed by the changes in the AS and AR grade severity between the last and baseline echocardiographic studies. In patients diagnosed with aortic stenosis at the baseline study (mean gradient > 10 mmHg), the annual increase rate of the mean gradient during follow-up was analyzed. If aortic surgery was performed, the last echocardiogram was the last test available prior to intervention. Aortic valve surgery indications were based on contemporary guidelines [16].

### 2.3. Statistical Analysis

Continuous demographic variables were expressed as mean SD. The Kolmogorov–Smirnov test was used to evaluate the normal distribution of variables. Intergroup differences for continuous parameters were assessed by Student’s *t*-test or ANOVA if they presented a normal distribution or analysis of variance with Bonferroni correction for multiple comparisons, and Mann–Whitney U test or Kruskal–Wallis if they did not present a normal distribution. For categorical variables, general characteristics of the sample were assessed by percentages (χ^2^ or Fisher exact tests). Logistic regression was used to identify independent variables associated with aortic root and ascending aorta dilations, and aortic stenosis and regurgitation progressions adjusted by follow-up time. Variables were entered in the model if *p* < 0.20 on univariate analysis. A two-tailed *p* value < 0.05 was considered statistically significant. STATA software version 15.1 was used for the analysis.

## 3. Results

### 3.1. Patient Baseline Characteristics, BAV Dysfunction and Aorta Dilation

A total of 718 consecutive patients (median age: 47.8 [IQR 33–62] years, range: 18–82; 69.2% male) fulfilled the inclusion criteria. Baseline characteristics are shown in Table 1. BAV-RL was significantly more frequent (81.5%) than BAV-RN (17.0%) and BAV-LN (1.5%). Pure BAV without raphe was observed in 11.3% of cases. Non-aorta dilation was present in 39 patients (29.3%), ascending aorta phenotype in 76 (57.1%), and root phenotype in 18 (13.5%). Aortic root diameter was greater in those with BAV-RL than RN and LN (*p* < 0.001); however, ascending aorta diameters showed no differences among valvular morphotypes.

In the baseline study, no or mild valvular dysfunction was present in 403 (56.1%) patients, moderate valvular stenosis in 116 (16.2%) and moderate regurgitation in 199 (27.7%). More-than-mild aortic calcification was present in 57 individuals (7.9%). Aortic valves with raphe were more frequently dysfunctional, with significant aortic stenosis in 18.2% vs. 0% (*p* < 0.001), significant regurgitation in 38.9% vs. 16.1% (*p* < 0.001) and calcification in 8.9% vs. 0% (*p* < 0.001).

### 3.2. Aorta Dilation and Valvular Dysfunction Progression

After a mean follow-up of 7.2 years [IQR 5–8] (range 5–15 years), aortic root diameters by TTE had progressed a mean of 0.23 ± 0.15 mm per year (Figure 1A). Greater annual aortic root dilation was significantly associated with male sex (0.24 ± 0.15 mm/year vs. 0.18 ± 0.14 mm/year; *p* < 0.0001), arterial hypertension (0.29 ± 0.14 mm/year vs. 0.20 ± 0.14 mm/year; *p* < 0.0001), smoking (0.25 ± 0.16 mm/year vs. 0.20 ± 0.15 mm/year; *p* = 0.02), presence of raphe (0.24 ± 0.15 mm/year vs. 0.15 ± 0.12 mm/year; *p* < 0.0001), valvular morphotype (BAV-RL: 0.23 ± 0.15 mm/year vs. BAV-RN: 0.19 ± 0.13 mm/year vs. BAV-LN: 0.19 ± 0.15 mm/year; *p* = 0.001), and significant aortic regurgitation (0.30 ± 0.14 mm/y vs. 0.18 ± 0.15 mm/year *p* < 0.0001) (Table 2) (Figure 2A). Variables related to rapid dilation of the aortic root (considered as an increase of ≥ 0.35 mm/year) are described in Table 3.

During the study period, 232 patients (68%) maintained a normally-functioning valve. AS grade severity progressed in 345 (48.1%), in 76 of them from moderate to severe (65.0%). AR progressed in 314 patients (43.3%), in 67 of whom, severity increased from moderate to severe (51.9%). Variables associated with AS progression adjusted by follow-up time were basal AS, presence of raphe, basal calcification, age, dyslipidemia, hypertension, diabetes and smoking (Table 4). Multivariate analysis showed basal AS severity, arterial hypertension, dyslipidemia, presence of raphe and follow-up period to be associated with AS progression.

Annual ascending aorta growth rate was 0.43 ± 0.32 mm/year (Figure 1B), being greater in patients with hypertension (0.56 ± 0.28 mm/year vs. 0.37 ± 0.32 mm/year *p* < 0.001) and in those with significant AS (0.50 ± 0.33 mm/year vs. 0.41 ± 0.32 mm/year *p* = 0.02) and AR (0.52 ± 0.4 mm/year vs. 0.46 ± 0.4 mm/year *p* = 0.02) (Figure 2B). Variables related to rapid dilation progression of this aortic segment (considered as a growth ≥ 0.7 mm/year) were described in Table 3.

Rapid annual progression of mean gradient > 2 mm/year in AS patients was related to raphe [OR:8.3(CI:1.12–61.55); *p* = 0.04], diabetes [OR: 2.6 (CI:1.25–5.46); *p* = 0.01], dyslipidemia [OR:1.8 (CI:1.11–2.95); *p* = 0.18], BAV-LN [OR:9.3 (CI:2.28–38.06); *p* = 0.02] and basal mean gradient [OR: 1.1 (CI: 1.08–1.13); *p* < 0.0001]

Variables associated with AR progression adjusted by follow-up time are specified in Table 4. Calcification of the valves increased in nearly half the cohort (41.4%). Valve calcification progression was greater in patients with hypertension (50.3% vs. 37.9% *p* = 0.004), diabetes (62.5% vs. 40.2% *p* = 0.008) and dyslipidemia (55.9% vs. 35.9% *p* < 0.001) and those with raphe (44.3% vs. 18.8% *p* < 0.001).

### 3.3. Clinical Follow-Up

During the follow-up period, 15.6% of patients required surgical treatment. The main reasons for surgical indication were: severe AS in 6.8%, ascending aorta enlargement in 4.9%, severe AR in 3.2% and aortic root dilation in 1%.

## 4. Discussion

The results of this study provide novel insights into the progression of aorta dilation and valvular dysfunction severity in a BAV population with no advanced disease. Our data suggested a clear relationship between arterial hypertension, raphe and valvular dysfunction with aorta dilation progression. In addition, arterial hypertension, dyslipidemia, raphe and basal valvular dysfunction were associated with progressive valvular dysfunction over a mid-long-term evolution.

The mean ascending aorta enlargement rate per year was overall low, twice more at tubular level (0.43 ± 0.32 mm/year) than at the sinuses of Valsalva (0.23 ± 0.15 mm/year) and was determined for different factors. Aortic root enlargement was associated with male gender, presence of raphe, arterial hypertension, AR and inversely associated with BAV-RN morphotype. Ascending aorta enlargement was associated with arterial hypertension but also with AS and inversely related to age.

Several cross-sectional BAV studies reported contradictory associations between aorta dilation and valvular dysfunction with clinical or echocardiographic variables [5,17]; however, aorta dilation and valvular dysfunction progression are little known and most studies had a follow-up period less than 5 years. Annual mean aortic dilatation rates were greater in earlier reports, with values ranging from 0.8 to 1.2 mm/year [18,19], whereas more recent studies found a rate between 0.36 and 0.45 mm/year [17,20,21], which is similar to our results.

The association between valve morphology and aortic dilatation rates was suggested and contradicted in previous studies and showed a wide dispersion of dilatation rates [6,21,22]. Our data showed that larger root diameters were found in male patients with BAV-RL and with arterial hypertension, concurring with those described by Della Corte et al. [22]. A higher frequency of aortic root dilation was described previously in men with BAV compared with women [6,23]. These sex differences, in addition to the higher prevalence of BAV in men, strongly suggest the causability of a genetic or embryologic factor. Regarding the association between aortic root enlargement and BAV-RL and the less aortic root dilation associated with BAV-RN, recent 4D-MRI studies showed that different BAV-phenotypes present different flow patterns with an anterior jet distribution in BAV-RL, whereas BAV-RN patients present a predominant posterior outflow jet at the upper part of the sinotubular junction that shifts to anterior or right anterior in mid and distal ascending aorta. Thus, BAV-RL patients present a higher axial WSS at the aortic root while BAV-RN present a higher circumferential WSS in the mid and distal ascending aorta [24]. These results may explain different aorta dilatation morphotypes in the BAV population. In our series the faster aortic root dilation was associated with male sex, hypertension, raphe and AR.

By contrast, ascending aorta enlargement at tubular level was related to arterial hypertension, aortic stenosis and younger age. Similar to our results, other studies showed a multivariate association between aorta growth rate and younger age [19,21,22]. Some studies found the aorta enlargement rate in AS to be twice as high at tubular level than at sinuses of Valsalva level in BAV and TAV [25]. We found the enlargement rate in the tubular part to be associated with valvular dysfunction but not determined by BAV morphotype.

Limited data are available regarding the progression of valvular dysfunction in BAV. While some cross-sectional studies in adults suggested no associations between BAV morphotypes and valvular dysfunction [26,27], the BAV-RN phenotype in children was linked with accelerated dysfunctional valve disease [28]. Adult patients with already mild-to-moderate AS have more rapid hemodynamic and valve degeneration progression compared to TAV patients of similar age and risk profile [26]. In our cohort, valve calcification and AS progression were associated with arterial hypertension and dyslipidemia, in addition to the follow-up period and presence of raphe. These results were in line with findings reported in other cross-sectional studies [10,17]. Yang et al. [11] suggested that rapid progression of the stenosis was determined by cardiac risk factors, particularly in BAV patients <60 years of age. Thus, strategies targeting strict primary prevention, even stricter than in the general population should be implemented in clinical practice during follow-up of this specific cohort. In our analysis, the rapid annual progression of the mean gradient was related to dyslipidemia, diabetes, BAV-LN morphotype, the presence of raphe and basal mean gradient. However, annual progression of the mean gradient is not linear and significantly related to the basal gradient [29]

Similarly, progression of AR, in addition to the follow-up interval, was associated with hypertension, dyslipidemia, presence of raphe and basal AR. Michelena et al. [7], in a community study, showed age, dyslipidemia and the presence of raphe to be associated with valve degeneration. Based on these findings, control of cardiovascular risk factors and some pharmacologic treatment such as statin therapy could be beneficial and reduce or delay degeneration of the BAV [30]. Mechanisms of progression to severe aortic dysfunction are poorly understood and merit future studies for the factors leading to this less frequent but major complication to be determined.

BAV may independently trigger the development of valvular degeneration and generate aorta dilation through the increase in wall shear stress of the eccentric aorta jet; however, the accumulation of cardiovascular risk factors plays a significant role in the progression of valvular dysfunction and aorta dilation [11]. Although BAV aortopathy is generally an indolent disease, with slow mean growth rates, novel imaging parameters such as 4D flow could be highly useful in individual risk assessment and also as a predictor of dilation location. Further stratification may become possible with the development of new imaging techniques.

### Limitations

Since this was not a population-based study; it suffered from the common bias of outpatient-based studies conducted at tertiary centers. The study was not designed to analyze the natural history of the disease and do not accurately represent the natural history of BAV with mild-moderate valvular or aorta involvement. We did not analyze patients who did not fulfil the criterion of having a second echocardiographic study after 5 years of follow-up. Therefore, we did not include patients lost to follow-up in the first 5 years of the study. However, no BAV patients had a severe valvular disease or an aorta diameter ≥50 mm. Few studies analyzed the medium-long-term progression of aorta dilation and valvular dysfunction in non-severe BAV disease. This was a retrospective, multicenter, electronic database-based design with all the inherent limitations this could entail. In order to minimize the inclusion of discordant values or cases with limited information, all the data were reviewed and analyzed by expert clinicians in the field.

It must also be pointed out that follow-up duration varied for each subject, since it was not possible to rule out differences in aortic or valvular dysfunction progression between patients with shorter compared with longer follow-up. We did not use a longitudinal linear mixed regression model that adequately model repeated measurements since the retrospective nature of the study and the different follow-up controls used by each center might have rendered this approach difficult. Additionally, propensity score could not be used.

The small number of patients in the LN group should also be acknowledged. In this setting, the results obtained in the secondary analysis should be treated with caution. Since this subtype of variant is usually excluded from studies, information on its prognosis is lacking. In our case, the inclusion of each type of BAV leaflet fusion in the overall analysis permitted easier generalization of the results. The present findings are in accordance with the latter and support the notion that BAV morphology-related hemodynamics may cause aorta dilatation directly, but do not rule out a role of underlying ontogenetic defects or an interplay between both.

## 5. Conclusions

In BAV patients, cardiovascular risk factors such as systemic arterial hypertension and dyslipidemia are related to ascending aorta dilation and aortic valve stenosis progression. Strict and early control and treatment of these modifiable risk factors should be mandatory in this cohort, even in younger patients.

In addition, intrinsic valvular characteristics, such as specific valve morphotypes and presence of raphe have also shown an association with the progression of valvular dysfunction and aorta dilation, of course taking into account the inherent limitations of our study-design. Being able to predict this progression will allow us to establish better clinical and imaging follow-up intervals, individualize patient life style and improve the timing of valvular and aortic surgery.

## Figures and Tables

**Figure 1 jcm-10-05264-f001:**
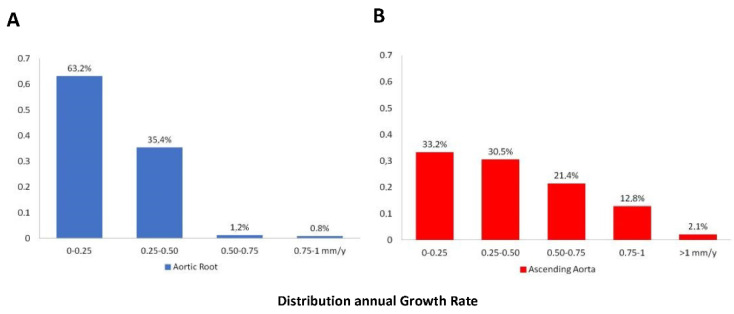
Histograms showing the distribution of mean annual growth rates of aortic root (**A**) and ascending aorta (**B**) diameters.

**Figure 2 jcm-10-05264-f002:**
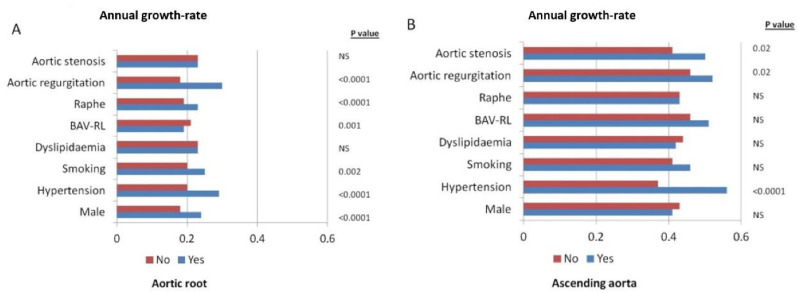
Aortic root (**A**) and ascending aorta (**B**) annual growth-rate diameters according to different baseline characteristics.

**Table 1 jcm-10-05264-t001:** Baseline characteristics of study participants according to BAV morphotype.

Variable	All Patients *n* = 718	BAV-RL *n* = 585 (81%)	BAV-RN *n* = 122 (17%)	BAV-LN *n* = 11 (2%)	*p* Value
Demographics and clinical data					
Age, years	47.9 ± 17.4	48.5 ± 17.6 ***	44.9 ± 16.5	46.6 ± 17.1	0.039
Male, *n* (%)	497 (69.2)	412 (70.4)	80 (65.6)	5 (45.5)	0.125
Smoking, *n* (%)	170 (23.6)	150 (25.6)	18 (14.8)	2 (18.2)	0.023
Hypertension, *n* (%)	200 (27.9)	167 (28.6)	30 (24.6)	3 (27.3)	0.670
Diabetes mellitus, *n* (%)	40 (5.6)	34 (5.8)	6 (4.9)	0 (0)	0.911
Dyslipidemia, *n* (%)	196 (27.3)	160 (27.4)	34 (28.1)	2 (18.2)	0.867
Valve abnormality and dysfunction					
Raphe, *n* (%)	637 (88.7)	524 (89.6)	102 (83.6)	11 (100)	0.096
Calcification > mild, *n* (%)	57 (7.9)	48 (8.2)	8 (6.6)	1 (9.1)	0.421
Normofuntional	341 (47.5%)	282 48.2%)	58 (47.5%)	1 (9.0%)	0.623
AS, *n* (%)	116 (16.2)	94 (16.1)	19 (15.6)	3 (27.3)	0.566
AR, *n* (%)	261(36.4)	209 (35.7)	45 (36.9)	7 (63.6)	0.184
Aortic diameter, dilation and morphotype					
Aortic root, mm	36.3 ± 5.4	36.9 ± 5.4 *	33.9 ± 4.2	32.9 ± 7.1	<0.001
Ascending aorta, mm	39.2 ± 6.2	39.2 ± 6.3	39.0 ± 5.7	40.7 ± 6.6	0.812
Sinusal Z score	1.3 ± 1.4	1.4 ± 1.4 *	0.8 ± 1.3	0.5 ± 1.9	<0.001
Ascending aorta Z score	2.8 ± 1.5	2.7 ± 1.5	2.9 ± 1.5	3.5 ± 1.7	0.176
Non-dilated aorta	181 (25.2)	148 (25.3)	31 (25.4)	2 (18.2)	0.953
Aortic root morphotype	86 (12.0)	84 (14.4) *	2 (1.6)	0 (0)	<0.001
Tubular morphotype	451 (62.8)	353 (60.3)	89 (72.9)	9 (81.8) ***	0.02

* *p* < 0.001; *** *p* < 0.05. AS: Aortic stenosis AR: Aortic regurgitation NS: not significant.

**Table 2 jcm-10-05264-t002:** Factors associated with annualized aortic diameter progression (mm/year) identified by univariate and multivariable linear regression analysis.

	Aortic Root				Ascending Aorta			
	Univariate Analysis Coef. (95% CI)	*p* Value	Multivariate Analysis Coef. (95% CI)	*p* Value	Univariate Analysis Coef. (95% CI)	*p* Value	Multivariate Analysis Coef. (95% CI)	*p* Value
Age	−0.0003 (−0.0009–0.0004)	0.407			−0.002 (−0.003–0.001)	0.006	−0.003 (−0.005–−0.002	<0.0001
Male sex	0.058 (0.034–0.082)	<0.0001	0.022 (0.001–0.044)	0.046	0.021 (−0.029 −0.072	0.419		
Hypertension	0.093 (0.069–0.117)	<0.0001	0.078 (0.056–0.099)	<0.0001	0.191 (0.141–0.242)	<0.0001	0.218 (0.167–0.268)	<0.0001
Smoking	0.031 (0.005–0.057)	0.021			0.045 (−0.011–0.099)	0.114		
Diabetes	−0.016 (−0.064–0.033)	0.524			0.052 (−0.111–0.094)	0.877		
Dyslipidemia	0.016 (−0.009–0.041)	0.197			0.012 (−0.041–0.065)	0.649		
Raphe	0.087 (0.052–0.122)	<0.0001	0.053 (0.021–0.084)	0.001	0.055 (−0.019–0.129)	0.145		
BAV-RL	−0.047 (−0.077–−0.017)	0.002	−0.041 (−0.067–−0.014		−0.032 (−0.101–0.024)	0.228		
Basal AS	0.005 (−0.026–0.035)	0.766			0.083 (0.019–0.146)	0.011	0.104 (0.042–0.166)	0.001
Basal AR	0.118 (0.097–0.139)	<0.0001	0.105 (0.084–0.126)	<0.0001	0.051 (0.021–0.081)	0.07		
z-score	0.024 (0.010–0.038)	<0.001			0.038(0.020–0.056)	<0.001		
Root morphotype	0.008 (0.004–0.0011)	<0.001			0.082(0.048–0.116)	0.09		
Tubular morphotype	0.090 (0.051–0.128)	0.142			0.005 (0.0002–0.009)	<0.001		

AS: Aortic stenosis AR: Aortic regurgitation BAV-RL: Bicuspid aortic valve right-left morphotype.

**Table 3 jcm-10-05264-t003:** Predictors of fast progressive aorta dilation.

	Aortic Root ≥ 0.35 mm/y				Ascending Aorta ≥ 0.70 mm/y			
	Univariate Analysis OR (95% CI)	*p* Value	Multivariate Analysis OR (95% CI)	*p* Value	Univariate Analysis OR (95% CI)	*p* Value	Multivariate Analysis OR (95% CI)	*p* Value
Age	0.990 (0.975–1.005)	0.190			0.988 (0.977–0.999)	0.034		
Male sex	2.041 (1.303–3.195)	0.002	2.112 (1.252–3.642)	0.007	1.094 (0.718–1.666)	0.676		
Hypertension	3.328 (2.269–4.879)	<0.001	2.705 (1.188–6.162)	0.018	2.344 (1.575–3.490)	<0.001	4.825 (2.185–10.652)	<0.001
Smoking	1.342 (0.887–2.032)	0.164			0.994 (0.633–1.561)	0.980		
Raphe	4.111 (1.630–10.362)	0.003	2.341 (1.223–4.483)	0.010	1.123 (0.599–2.104)	0.716		
BAV-RL	0.559 (0.319–0.982)	0.043			0.949 (0.567–1.588)	0.841		
Basal AS	1.002 (0.608–1.649)	0.995			1.613 (1.002–2.599)	0.049		
Basal AR	4.784 (3.234–7.078)	<0.001	9.936 (3.051–32.350)	<0.001	1.251 (0.721–2.341)	0.081		

AS: Aortic stenosis AR: Aortic regurgitation BAV-RL: Bicuspid aortic valve right-left morphotype.

**Table 4 jcm-10-05264-t004:** Univariate and multivariate analysis of AS and AR adjusted by follow-up time.

	AS Progression				AR Progression			
	Univariate analysis OR (95% CI)	*p* Value	Multivariate Analysis OR (95% CI)	*p* Value	Univariate Analysis OR (95% CI)	*p* Value	Multivariate Analysis OR (95% CI)	*p* Value
Male sex	1.099 (0.795–1.521)	0.566			1.521 (1.090–2.121)	0.014		
Hypertension	1.736 (1.236–2.441)	0.001	1.553 (1.079–2.237)	0.018	5.549 (3.831–8.039)	<0.0001	5.372 (3.651–7.904)	<0.0001
Smoking	1.212 (0.849–1.728)	0.289			1.632 (0.143–2.33)	0.007		
Diabetes	2.526 (1.225–5.211)	0.012			1.052 (0.545–2.029)	0.880		
Dyslipidemia	2.461 (1.740–3.482)	<0.0001	1.709 (1.64–2.509)	0.006	2.726 (1.929–3.851)	<0.0001	2.292 (1.576–3.332)	<0.0001
Raphe	6.226 (3.267–11.864)	<0.0001			4.083 (2.221–7.503)	<0.0001	3.558 (1.859–6.810)	<0.0001
BAV-RL	1.083 (0.728–1.612)	0.692			0.934 (0.625–1.398)	0.742		
Basal AS/AR	2.461 (1.613–3.754)	<0.0001	1.621 (1.034–2.542)	0.035	1.357 (0.991–1.858)	0.057	1.433 (1.062–1.217)	<0.0001
Valvular calcification	1.960 (1.109–3.462)	0.02						

AS: Aortic stenosis AR: Aortic regurgitation BAV-RL: Bicuspid aortic valve right-left morphotype.

## Data Availability

The data presented in this study are available on request from the corresponding author.

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
