# Peer review of "Predictors of Ascending Aorta Enlargement and Valvular Dysfunction Progression in Patients with Bicuspid Aortic Valve"

_jcm, 2021, doi:10.3390/jcm10225264_

Round 1
Reviewer 1 Report
In the proposed paper: Predictors of ascending aorta enlargement and valvular dysfunction progression in patients with bicuspid aortic valve the authors performed a retrospective analysis on echocardiographic data in 718 patients at 10 different centers. To get insights into the dynamics of ascending aorta dilatation, progression of Aortic valve stenosis and insuficceny is highly interesting for a better understanding of the disease and patient management.
1.) Echocardiography is an examiner-dependent method. How you ensured that the examinations were evaluated in a comparable manner? Was there a core lab? Was there an SOP for the measurements? I recommend to stress this topic a bit more in detail in the paper.
2.) As this is a retrospective study, I assume that there was not a defined follow up interval. In what intervals the patients had a echocardiographic follow up? How many follow up visits a patient had in avarage? And what was the mean time interval between the visits?
3.) CT Angiography is still the gold standard for measurements on the Aorta. Did the patients in yout study only received TTE examinations or you also have CTA examinations available. If yes it will be quite interesting to compare the size measurements.
Author Response
Thank you very much for your comments concerning our recently re-submitted manuscript. We have revised carefully the referee #1 and #3 concerns, providing a detailed response to their comments.
Reviewer 1#
In the proposed paper: Predictors of ascending aorta enlargement and valvular dysfunction progression in patients with bicuspid aortic valve the authors performed a retrospective analysis on echocardiographic data in 718 patients at 10 different centers. To get insights into the dynamics of ascending aorta dilatation, progression of Aortic valve stenosis and insuficceny is highly interesting for a better understanding of the disease and patient management.
Echocardiography is an examiner-dependent method. How you ensured that the examinations were evaluated in a comparable manner? Was there a core lab? Was there an SOP for the measurements? I recommend to stress this topic a bit more in detail in the paper.
Thank you for your suggestion. There was not a core lab to review the images. All of the studies had been made in centers with experienced echocardiographists who had made the measurements following international recommendations of the American Society of Echocardiography and EACVI as we point in the methods section. In addition, our group has also taken part of previous studies such us BICATOR trial and others registries (Heart. 2018 Apr;104(7):566-573). Following your recommendation we have included also this information in methods.
As this is a retrospective study, I assume that there was not a defined follow up interval. In what intervals the patients had a echocardiographic follow up? How many follow up visits a patient had in avarage? And what was the mean time interval between the visits?
Given the retrospective nature of this study, it is correct that there was not a defined follow up interval. Considering that our cohort includes patients with no significant valvulopathy or aortopathy, patients were evaluated every 2-3 years (mean 2.6 years) following clinics criteria
CT Angiography is still the gold standard for measurements on the Aorta. Did the patients in yout study only received TTE examinations or you also have CTA examinations available. If yes it will be quite interesting to compare the size measurements.
We appreciate your comment. In our study, and also due to the lack of severity, CTA were not performed systematically. In this sense, and although some patients do have CTA, this exploration was not mandatory and it did not was part of our data.
Reviewer 2 Report
This is a retrospective observational study that includes a commendable number of subjects and a follow-up that is >5 years. Even though some aspects of the design of the study can be criticised as well as the fact that echographic data were taken from 10 tertiary hospitals- which the authors acknowledge as a limitation- I consider this paper a worthwhile addition to the knowledge of bicuspid aortopathy progression in adults.
Author Response
Thank you so much for your comments. We appreciate your review.
Reviewer 3 Report
A very interesting study that focuses on very important issues.
The authors do not provide a list of monitored parameters in the methodology of the work.
I observe the greatest insufficiency in the group of patients, which is very diverse. Patients are included in the follow-up based on a visit to the cardio center for an unidentifiable reason. The study thus inadvertently selects inclusion criteria.
The conclusion of the work very modestly evaluates the results, which is correct. However, the work in this wording can be quite confusing for the less interested reader.
I recommend the authors to consider the use of "propensity score matching" at least for the L-R group. In case the authors evaluate that the use is not possible, this fact, as well as the possible influence of the conclusions must be emphasized both in the conclusion and in the abstract of the work.
Author Response
The authors do not provide a list of monitored parameters in the methodology of the work. I observe the greatest insufficiency in the group of patients, which is very diverse. Patients are included in the follow-up based on a visit to the cardio center for an unidentifiable reason. The study thus inadvertently selects inclusion criteria.
Thank you so much for your comments. Monitored parameters are the already described in the methods section. The only quantitative variable not indicated is the aortic mean gradient. Following the last guidelines of the ASE and EACVI, the severity of the valvulopathy was defined after the integration of the different parameters measured (semiquantitative and quantitative). We are afraid that it is not a population-based or epidemiological study like those of Michelena et al, with its intrinsic limitations.
The conclusion of the work very modestly evaluates the results, which is correct. However, the work in this wording can be quite confusing for the less interested reader.
According to your suggestion, we have made modified the conclusion.
I recommend the authors to consider the use of "propensity score matching" at least for the L-R group. In case the authors evaluate that the use is not possible, this fact, as well as the possible influence of the conclusions must be emphasized both in the conclusion and in the abstract of the work.
We appreciate your comment. We are afraid that in our case, propensity score use is not possible. We will remark it in the limitations section, but also in the conclusion and abstract. Despite this, the study provides interesting findings that should be analysed in new prospective longitudinal studies.